# Some Insights into the Inventiveness of Dinoflagellates: Coming Back to the Cell Biology of These Protists

**DOI:** 10.3390/microorganisms13050969

**Published:** 2025-04-24

**Authors:** Marie-Odile Soyer-Gobillard

**Affiliations:** CNRS-Sorbonne University, Oceanological Observatory, Laboratoire Arago, F-66650 Banyuls-sur-Mer, France; elido66@orange.fr; Tel.: +33-611895084

**Keywords:** dinoflagellates, cell biology, innovative features, evolution

## Abstract

In this review dedicated to the great protistologist Edouard Chatton (1883–1947), I wanted to highlight the originality and remarkable diversity of some dinoflagellate protists through the lens of cell biology. Their fossilized traces date back to more than 538 million years (Phanerozoic eon). However, they may be much older because acritarchs from the (Meso) Proterozoic era (1500 million years ago) could be their most primitive ancestors. Here, I described several representative examples of the various lifestyles of free-living (the autotrophic thecate *Prorocentrum micans* Ehrenberg and the heterotrophic athecate *Noctiluca scintillans* McCartney and other “pseudo-noctilucidae”, as well as the thecate *Crypthecodinium cohnii* Biecheler) and of parasitic dinoflagellates (the mixotroph *Syndinium* Chatton). Then, I compared the different dinoflagellate mitotic systems and reported observations on the eyespot (ocelloid), an organelle that is present in the binucleated *Glenodinium foliaceum* Stein and in some Warnowiidae dinoflagellates and can be considered an evolutionary marker. The diversity and innovations observed in mitosis, meiosis, reproduction, sexuality, cell cycle, locomotion, and nutrition allow us to affirm that dinoflagellates are among the most innovative unicells in the Kingdom Protista.

## 1. Preamble

This review is dedicated to the great protistologist Edouard Chatton (1883–1947), a pioneer in the study of the cytology of many (free-living and mixotrophic) dinoflagellates, who revealed the complexity and originality of these protists at the beginning of the 20th century. E. Chatton (1883–1947) was the first scientist to take a detailed interest in the description of the external morphology of many dinoflagellates and, especially, in their life cycle, cytology and mitosis (dinomitosis and syndinian mitosis). He masterfully described his findings in his PhD thesis, published more than one hundred years ago [1]. Chatton reported all his observations in his monumental *Titres et Travaux Scientifiques* (in English, *Titles and Scientific Works*) [2], a publication enriched by splendid course boards (160/110 cm), genuine artistic works he drew for his students, starting in 1920. A selection of the finest drawings by Chatton have been recently published in a book [3]. Chatton was also the first to distinguish between prokaryotic and eukaryotic microorganisms [4].

## 2. Introduction

Opening the chapter on the Phylum Dinoflagellates, written by FJR Taylor in the 1990s [5], is enough to appreciate the immense diversity of these protists. The current classification recognizes ~550 genera and a total of 6000 described species that play a prominent role in water ecology. Molecular clock and biogeochemical indices indicate that the dinoflagellate lineage diverged ~650 Ma, and fossil traces of thecate cells and of cysts/zygotes appeared during the Triassic [6]. Evidences of dinoflagellate existence come from the biogeochemical analysis of early Cambrian sediments (~520 million years ago), in which dinoflagellate-specific sterols (i.e., dinosterols) were detected in rock extracts and petroleum. However, some scholars think that they may have an older, earlier Precambrian origin, forming part of the Acritarchs [6,7]. According to B. Dale (2023) [8], the first dinoflagellates would have been naked and, therefore, without fossil traces except for biogeochemical residues. Only later they would acquire a protective theca to escape various predators. According to the phylogeny analyses by Bachvaroff et al. [9] and by Delwiche [10], they belong to the alveolates and can be autotrophic (about half are photosynthetic), heterotrophic, parasitic, and/or mixotrophic, symbiotic, plasmodial, or organized as “bi-nucleated” or “pseudo-multinucleated” cells. They all live in aquatic environments (seawater, brackish, or fresh water), and they have hemolymph, gut, or coelomic cavity for mixotrophic dinoflagellates [11]. The composition of their chromatin and the organization of their nucleus (dinokaryon) are also unique, as well as their “dinomitosis” [6,12,13]. In the nucleus of many dinoflagellate species, chromosomes are maintained in a quasi-permanent, compacted state. Molecular studies to estimate their diversity and to attempt their classification were carried out, especially to locate dinoflagellates among other protists and to classify dinoflagellate taxa [14]. Phylogenetic trees were built using protein sequences (e.g., dimeric iron-containing superoxide dismutases) [15] or the sequence of specific genes (e.g., 18S rRNA, 5.8S rRNA, 24S rRNA, or SS rRNA). All genomic data have been regrouped in the recent and very complete review by Serjie Lin [16].

In this review, I chose to present several examples of different categories of these original protists, from a non-exhaustive list of representatives of their different lifestyles, in order to highlight their remarkable diversity and adaptability, but without classifying them from the simplest to the highest evolutive complexity. I then focused on some innovative features, several of which were previously but partially described [6] using cell biology tools. I particularly highlight the acquisition of an eyespot (ocelloid) in several species belonging to the Warnowiidae family.

## 3. Materials and Methods

A review of the literature was carried out by searching the PubMed and Google Scholar databases from 2000 to 2024 following the preferred reporting items for systematic reviews and meta-analyses (PRISMA) guidelines, except for pioneering references (from 1920 to 2000). Studies were identified using the following keywords: dinoflagellates, eyespot, mitotic apparatus, *Prorocentrum*, *Noctiluca*, *Crypthecodinium, Syndinium.* Many illustrations in this review were taken from the selected publications, after having obtained permission from the publishers. For unpublished microphotographs taken by the author, the preparation/analysis methods were described in [17,18].

## 4. Results

This review summarizes the amazing and innovative diversity of dinoflagellates (autotrophic, heterotrophic, or mixotrophic) and their adaptations.

### 4.1. An Autotrophic Free-Living Dinoflagellate: Prorocentrum micans Ehrenberg

#### 4.1.1. General Features

The cell cycle of *P. micans* Ehrenberg vegetative cells (Figure 1a’) was deciphered by Bhaud and Soyer-Gobillard [19], and it lasts 6 days, which is a relatively long cell cycle. This protist is characterized by an original flagellar system: an undulating membrane attached to the polysaccharidic epitheca that ends in a short transverse flagellum and a longitudinal flagellum (Figure 1a–c).

#### 4.1.2. Original Features

##### Nucleus

The nucleus is filled with a hundred large chromosomes (10 µm long and 1 µm wide) (Figure 2**a**) and surrounded by large plastids, not visible in this nucleus picture. As these chromosomes are quite large, Haapala and Soyer-(Gobillard) could observe them using transmission electron microscopy (TEM) for the first time after spreading them on water (Figure 2**b**). This allowed for describing their organization, particularly the superhelical twisted structure of their nucleofilaments [22]. The twisted and regular unwinding of these chromosomes led to the hypothesis that they are organized in multiple circular chromatids (Figure 3**c**) [22,23]. This hypothesis was later confirmed and thoroughly described by Oakley and Dodge [24]. The isolation of chromatids from purified DNA molecules, spread and shadowed with platinum carbon, was particularly difficult to achieve [25] (Figure 2**c**).

*P. micans* has two types of reproduction: a vegetative reproduction, carried out by binary fission, and a sexual one. In a vegetative cell, dividing chromosomes must be attached to the internal part of the nuclear envelope (Figure 3**a**,**b**) and are not directly in contact with the microtubular spindle located in cytoplasmic channels that pass through the mitotic nucleus. This observation is supported by the absence of centromeric heterochromatin in *P. micans* [26]. In these rod-shaped chromosomes, centromeres could be localized at the chromosome tip and can be considered to be (virtual) telomeres. However, due to their lack of centromeric heterochromatin, *P. micans* could be considered as a kind of “immortal cell”, unable to undergo apoptosis.

*P. micans* chromosomes are devoid of histones and nucleosomes but are rich in 5-hydroxymethyluracil [27]. However, nucleosomal structures could be easily reconstructed in vitro using *P. micans* DNA and purified corn histones, showing that the presence of this abnormal base is not an impediment to their reconstruction [28]. *P. micans* also allowed us to carry out some of the first phylogenetic molecular analyses using ribonucleoprotein subunit (SSU, LSU) sequencing data [29].

Interestingly, using immunogold electron microscopy after sample preparation by vitrification at the liquid helium temperature (−269 °C) [17], a technique developed by Professor Jacques Dubochet (Nobel Prize in Chemistry) that preserves the antigenic sites particularly well, we could detect two DNA types in *P. micans* chromosomes: B-DNA (right-handed DNA) and the so-called “mirror” Z-DNA (left-handed DNA). B-DNA represents the usual DNA conformation (right-handed double helix). It is the most represented in the living world (Figure 3f), unlike Z-DNA (which turns left). Thanks to work in collaboration with Dr Etienne Delain’s group at the Gustave Roussy Cancer Research Institute in Villejuif (France), who produced an anti-Z-DNA antibody from urine samples of people with cancer [18], we could detect the presence of Z-DNA as clusters of gold particles at the chromosome periphery (Figure 3f) and in the chromosomal fission zones (Figure 3**e**). Conversely, B-DNA was localized in the chromosome body (Figure 3d,f). We hypothesized that in these always compacted chromosomes, transcription takes place at the chromosome periphery and DNA opening occurs through a loop opening process dependent on the mirror Z-DNA [18].

##### Sexual Reproduction

*P. micans* sexual reproduction was observed in our laboratory by chance, after a glass bottle with some cultured cells was placed in a refrigerator at 4° in the dark for 12 h. The next morning, many vegetative cells (n = 1qDNA) had paired up, clinging to one another through their apical spine and emitting a connecting tube (i.e., fertilization tube) (Figure 4**A**–**C**) through which one cell (male?) injects its genetic material surrounded with nuclear envelope into the other (the female?) (Figure 4**D**,**E**) [29]. Shortly after the fusion of the two nuclear envelopes and nuclei, their chromatin undergoes a very impressive circular and rapid movement on the right named chromatic cyclosis as observed with phase contrast optical microscope for several minutes, during which the male and female genetic materials mingle and chromosomes completely lose their regular helical structure (Figure 4**F,f,f’**). By quantifying DNA in single cells, we showed that in *P. micans*, early planozygotes (Figure 4**f**) (n = 2qDNA content) double their DNA to n = 4qDNA before the first of the two zygotic divisions (meiosis), which leads to n = 1qDNA in vegetative cells. We also observed that nuclear chromatic cyclosis occurred just before the first meiotic division and that the chromosome nucleofilament organization was altered [21]. Nuclear chromatic cyclosis was discovered in two *Ceratium* species by Pouchet in 1885 [30] and thoroughly described in six *Peridinium* species by Biecheler in 1935 [31]. Many other dinoflagellates present both vegetative (binary reproduction) and sexual reproduction, the last occurring often when the physicochemical conditions are unfavorable.

##### Intriguing Intranuclear Supercoiled Microcables During Meiosis [32]

During meiosis, TEM observations showed a total disorganization of the chromosomal nucleofilaments. Specifically, the chromosomes are very uncoiled but still well individualized (Figure 4**f’**) [29] to allow for intergenomic exchanges. During this period, chromatic cyclosis occurs, and microcables are visible in the nucleoplasm from TEM observations (Figure 5**a**) [32]. They are composed of twisted filaments organized in a tetrahelix in which the unitary microfilaments have a diameter of 60 Å. They are organized in a right tetrahelix with a diameter that varies from 35 to 45 nm (Figure 5**a’,b,c**). The unit microfilament diameter corresponds to that of actin that we detected and described for the first time in a dinoflagellate [33]. The presence of actin was confirmed by Berdieva et al. (2018) [34] who demonstrated the presence of F-actin in the cytoplasm and G-actin in the nucleus at the nucleolar level in the dinoflagellate *Prorocentrum minimum* (Pavillard) Schiller. This evidence allowed us to hypothesize that microcables could be the driving element of chromatid mixing during chromatic cyclosis as well as of chromosome transport during dinomitosis.

### 4.2. Two Heterotrophic Free-Living Dinoflagellates: Noctiluca Scintillans Mc Cartney and Crypthecodinium cohnii Biecheler

#### 4.2.1. *Noctiluca Scintillans* McCartney

##### General Features

Commonly called “Fire of the Sea”, *N. scintillans* McCartney is a non-parasitic athecate free-living spherical protist (Figure 6**a**–**c**) without plastids that lives in marine environments. It exhibits bioluminescence when agitated, through a luciferin-luciferase system in its cytoplasm [35,36], like some other dinoflagellate species. These planktonic dinoflagellates, which have not been recorded to produce toxins, can bloom into non-toxic “red tides” that spread as a several-centimeter-thick layer on the sea surface. *N. scintillans* McCartney does not have cellulose thecal plates, but it is not completely “naked”. Indeed, its cell covering or “amphiesma” presents an outer membrane that surrounds the cell, amphiesmal vesicles, and a more or less thin pellicular layer, the epiplasm, as described by Melkonian and Höhfeld [37]. *N. scintillans* McCartney can divide by binary fission after loss of the tentacle or by sporulation after sexual reproduction [38] to form sporocytes (Figure 6**a**–**c**) and then uniflagellate spores, which will ensure the dissemination and resistance to growth-unfavorable conditions (Figure 6**b**) [39,40,41]. During sporogenesis, their chromatin progressively condenses as the divisions progress [42]. Chromosomes are well individualized only in mature spores where there are at least 120 chromosomes per nucleus.

##### Original Features

In the trophozoite and at the start of divisions, the inner part of the nuclear envelope is lined in a peripheral fibrous zone (Figure 7**a,b**) with nuclear ampullae that bear nuclear pores and actively participate in sporogenesis preparation (**Figure 7a**–**d**). After the calculation, the surface area of the ampullae is twice the surface area of the total nuclear envelope [40]. Trophozoites can undergo a binary division or form spores. In a very short time, nuclear ampullae constitute nuclear membrane reserves immediately available during sporulation, accelerating the divisions during a bloom (red tide). Indeed, during the beginning of sporulation, nuclear ampullae fuse to leave a smooth nuclear envelope devoid of ampullae in the spores (Figure 8**b**) [40,41]. Sporulation is triggered by physicochemical reasons not yet specified in the surrounding environment, and then, multiplication is very rapid by splitting of the chromatic mass, toward stage 16 nuclei during which chromatin condenses and individualizes chromosomes (Figure 9).

Cytochemical investigations did not detect histone-like proteins in the trophozoite nucleus. At the molecular level, the unusual 5-hydroxymethyluracil base has been found in the nuclear DNA of *N. scintillans* [27]. In the investigated dinoflagellate species, this unusual base replaces 12–68% of all thymine residues, a relatively important replacement amount.

Cytoskeletal elements that participate in *N. scintillans* trophozoite motility are also involved in nutritional functions. They are located in the tentacle (Figure 10**A,B**) and at the level of the oral apparatus (cytostome) where cytoplasmic myofibrils are organized in striated and contractile fibers named “myonemes” (Figure 11**a,b,b’**). We were the first to observe these striated fibrillar structures in *N. scintillans*, which are similar to the striated myonemes in its contractile tentacle [39,43]. Myonemes are inserted on the epiplasmic membrane (E), equipped with a large number of microtubule rows (Figure 11**D**, arrowheads) that are crosslinked with each other (Figure 11**D**, arrows) [39]. Striated myonemes are distributed along the tentacle and linked to one another by an axial knot (Figure 11**b,c**). Tentacle contraction involves epiplasmic membrane deformation, myoneme contractility, and microtubule modifications [43].

The oral apparatus (cytostome) opening and closing (Figure 12**a–c**) during prey capture is ensured by a complex system of curtains of striated (contractile) myofibers (Figure 12**d**) that run between the cytostome and are anchored at a reinforced furrow or “sulcus” (schematized in Figure 12**a,e**) [39]. Preys include pollen grains, other protists, and sometimes even congeners. It is amusing to note that this is one of the first recorded cases of cannibalism in the Kingdom Protista.

Phylogenetic analyses of this complex cell, using the gene sequences of beta-tubulin, HSP90 [44] or nuclear 28S rDNA [45], support its placement among dinoflagellates, not far from *Oxyrrhis marina* Dujardin. These species do not have a theca and probably appeared very early in the Precambrian era, according to Dale [8].

#### 4.2.2. Other Free-Living Bioluminescent Dinoflagellates: An Homage to Edouard Chatton’s Scientific and Artistic Talents [3].

E. Chatton (1930) represented these species on a magnificent course board (160/110 cm) drawn with colored pastels on black paper (Figure 13), for his students [3].

#### 4.2.3. *Crypthecodinium cohnii* Biecheler

##### General Features

The heterotrophic dinoflagellate *C. cohnii* Biecheler is a particularly fascinating protist despite its classical biflagellate morphology (Figure 12**a,b**). Kubai and Ris described its mitosis for the first time [46].

##### Innovative Features

*C. cohnii* presents a complex cell cycle [47]. One vegetative cell can perform two successive complete cell cycles (16 h), as shown in Figure 14C, and release four daughter cells. One of these new swimming cells releases two daughter cells 10 h later (external circle of the diagram). During this time, other swimming cells could produce two or four daughter cells. Different diagrams could be possible with different cycle lengths and numbers of daughter cells. Immunocytochemistry and confocal microscopy analyses of the microtubular cortex and mitotic apparatus [48,49,50] showed a well-developed microtubular cytoskeleton (Figure 15). Interestingly, we have shown that a HSP70-related protein is associated with the centrosome and is conserved from *C. cohnii* to human cells [51]. *C. cohnii* is rich in lipids [52], and among the heterotrophic marine dinoflagellates, is a prolific producer of docosahexaenoic acid (DHA), an important fatty acid [53]. Amusingly, after the publication of the study on the complex *C. cohnii* cell cycle by our laboratory [47], we were contacted on several occasions to give advice concerning the specific culture of this species by industrial groups in the USA working on the elaboration of artificial “maternal” milk or of DHA production.

### 4.3. A Mixotrophic Dinoflagellate: Syndinium spp. Chatton

#### 4.3.1. General Features

According to the first descriptions by Chatton (1910), the main hosts of this parasitic plasmodial dinoflagellate are the pelagic copepods of the Mediterranean Sea close to Banyuls-sur-Mer, France [1,2]. The “plasmodium”, which is composed of a cytoplasmic mass with numerous nuclei containing five V-shaped chromosomes and without cellular partitioning, is constantly in mitosis, growing in the general cavity (coelomic cavity) of copepods and other crustaceans and rapidly destroying all their vital organs. The mature biflagellate dinospores, totally devoid of plastids, are released into the seawater and very rapidly swim to quickly parasitize another prey.

#### 4.3.2. Innovative Features

Like *Hematodinium perezi* (Syndiniales) that was discovered by Chatton and Poisson [54] on the French coasts and parasitizes the blood of crabs, *Syndinium* sp. can also parasitize the general cavity of many crustaceans, especially copepods. Currently, these Syndiniales are widespread throughout the world, and many scientists have been working to refine their molecular characterization [55,56] in connection with important economic issues because they often infest edible crustaceans. Their destructive action due to their rapid multiplication can cause the death of many crustacean species in a short time, and their power of contamination by dinospores (totally devoid of plastids) is infinite.

Chatton considered syndinian mitosis as a particular mitosis in dinoflagellates [57] and described it in detail (Figure 16**a,b**) [1,2] in several different species that parasitize various copepod crustaceans or radiolarians. Subsequently, TEM observations by Ris and Kubai (1974) showed the originality of this mitotic system [58]. The compacted chromosomes are attached to the extranuclear microtubular mitotic spindle through the nuclear membrane by means of kinetochores via a large cytoplasmic channel [58,59]. They are connected to the centrosome region that contains two centrioles (Figure 17**a**) and to microtubules of the mitotic spindle. It is important to note that kinetochores appeared for the first time within dinoflagellates in the kingdom Protista. Later, when mature dinospores totally devoid of plastids are released into seawater (Figure 17**b**), their chromatin is totally fragmented.

In his different descriptions of *Syndinium* sp., Chatton observed three different spore types, with different sizes (Figure 15**B**) [2], that parasitize two different copepod species (*Clausocalanus arcuicornis* and *Paracalanus parvus*) and wondered about their roles. Later, Skovgaard et al. [55], using full-length SSU rDNA sequences of *Syndinium turbo* from these two copepod hosts, concluded that the three spore morphotypes are 100% identical and belong to a single species, *Syndinium turbo* Chatton. Moreover, phylogenetic analyses place *Syndinium* as a sister taxon of *Hematodinium* sp., a blue crab parasite according to Chatton and Poisson. The main innovations in *Syndinium* sp. essentially concern motility, nutrition, and reproduction (sexual or not), and also the appearance of histone-like basic nuclear proteins linked to their DNA [58,59], as well as the presence of kinetochores during mitosis [58].

## 5. Dinoflagellate Mitotic Apparatus as an Evolutionary Marker

Despite the great dinoflagellate diversity in terms of innovations, physiology, lifestyle, and cell cycle, they have a remarkably homogeneous mitosis mechanism, except for *Syndinium* spp., *Oxyrrhis marina,* and some *Amoebophrya* species. The system of cytoplasmic channels passing through the intact nucleus indicates that microtubules are never in direct contact with the chromosomes but are always separated from them by the persistent nuclear envelope, as shown in the “models” *C. cohnii* Biecheler and *P. micans* Ehrenberg (Figure 18) [49].

It should be noted that the kinetochores (contact between chromosomes and mitotic spindle) and the polarized centrosomes, which contain two centrioles, were first observed in Syndinidae. This kind of mitosis has been particularly well studied by Ris and Kubai (Figure 19**a**–**f**) [58].

From all the precedent data and their own findings in *Amoebophrya* spp., Moon et al. (2015) observed that not all species classified as dinoflagellates have an extranuclear spindle [60]. For instance, in some *Amoebophrya* spp. species, an extranuclear microtubule cylinder, located in a depression of the nuclear surface during the interphase, moves into the nucleoplasm via sequential membrane fusion events and develops into an entirely intranuclear spindle [60]. Their results suggest that the intranuclear spindle of *Amoebophrya* spp. may have evolved from an ancestral extranuclear spindle, as shown on the phylogenetic tree of different mitotic apparatuses in Figure 20.

## 6. A Rather Perfected System: The Eyespot (Ocelloid) of Dinoflagellates

Surprisingly, the eyespot (ocelloid) is one of the most evolved photosensitive organelles in protists [5] and is considered an important phylogenetic marker in dinoflagellates. The ocelloid is present in several heterotrophic athecate dinoflagellates from the *Warnowiaceae* family, such as *Nematodinium*, *Warnowia*, *Erythropsis*, in several *Woloszynskioids*, and in the autotrophic *Glenodinium* (*Peridinium*) *foliaceum* Stein, a binucleate dinoflagellate, with both a dinokaryon and another nucleus probably of endosymbiotic (diatom) origin. Greuet (1965) published the first description of an eyespot in the dinoflagellate *Erythropsis pavillardi* Hertwig [61], as reported by Gehring (2001) [62] and then Francis (1967) in *Nematodinium* spp. [63]. Using TEM, Greuet showed that this most sophisticated structure is ~25 µm long and 15 µm wide [64,65,66]. Its main characteristic is the presence of a transparent and domed hyalosome, which plays the role of the lens, and of a pigment layer, which plays the role of the retina (Figure 21**A,B**). This complex photosystem was observed in many species of the family of heterotrophic *Warnowiidae* [67,68] and also in the binucleated dinoflagellate *Glenodinium foliaceum* Stein, now *Kryptoperidinium triquetrum* (Ehrenberg) U. Tillmann, M. Gottschling, M. Elbrächter, W.-H. Kusber, and M. Hoppenrath, 2019. Its light transmission mechanism was elucidated by Kreimer [69].

In 1999, Kreimer [69] described the eyespot of the binucleated *G. foliaceum* in which two DNA types from two nuclei (a dinokaryon and a nucleus of endosymbiotic origin of diatom type) and two chloroplast types of the same origin have been detected [70]. In this dinoflagellate, the eyespot is located in the posterior part of the cell close to the sulcus (Figure 22**a**). It is composed of a pigment cup, retinoid, and lens and functions as a photoreceptor through which light can pass and be reflected outside (Figure 22**A**), or it can pass through the protist body before being reflected back out (Figure 22**B**), which determines the orientation of the cell swimming.

The eyespot of Warnowiidae is located in the anterior part of the cell. In *Erythropsidinium* spp., a heterotrophic dinoflagellate with a posterior appendage called a piston that plays a role in its locomotion, the eyespot occupies a significant volume of the cell (Figure 23**A**). Walter Gehring^†^ (1939–2014), to whom I would like to pay special tribute here, worked on the genetic control of eye development and the evolution of eyes and photoreceptors in the animal kingdom [62,71]. From the work of Greuet [61,64,65,66], he thought that the dinoflagellate ocelloid represents an evolutionary enigma because it looks like a multicellular camera-type eye, but is found in a unicellular protist. Then, in 2015, Hayakawa et al., with Walter Gehring [72] used TEM to determine whether the dinoflagellate ocelloid is functionally photoreceptive. They found that this sophisticated structure is composed of a retina and lens-like structures called the retinal body and of a transparent hyalosome (the lens) (Figure 23**A,B**). Moreover, they observed that the retinal body changes its morphology depending on the illumination conditions and that the hyalosome displays a refractile nature (Figure 24). Lastly, they identified a rhodopsin gene fragment by in situ hybridization in *Erythropsidinium* expressed sequence tags (ESTs) that is expressed in the retinal body [72] and is most closely related to bacterial rhodopsin. Therefore, with Gavelis et al. [73], they could strongly affirm that “Eye-like ocelloids of dinoflagellates are built from different endosymbiotically acquired components”.

## 7. Conclusions

As reported by Hayakawa et al. (2014) [72], Darwin wrote in his work “On the Origin of Species” [74] that the eyes are an example of organs of extreme perfection and complication. Darwin was convinced that they had only appeared thanks to natural selection. In the case of the *Erythropsidinium* ocelloid, a highly elaborate camera-type eye that resembles part of the metazoan eye [72,73] has evolved in a single cell, a protist dinoflagellate, and is probably the vestige of endosymbiosis. Despite the great diversity of these protists in terms of taxomony and innovations and the fact that some specific proteins associated with the centrosome are conserved up to human cells (e.g., HSP70-related protein) [51], and the presence of right (B-) and left (Z-) handed DNA in their chromosomes [18], we observe the result of their evolution as it appears today, after more than 1500 million years since the beginning, probably dating back to the Proterozoic era according to Brian Dale [8]. Much research using the most advanced techniques in cell biology is still necessary to try to solve these enigmas.

## Figures and Tables

**Figure 1 microorganisms-13-00969-f001:**
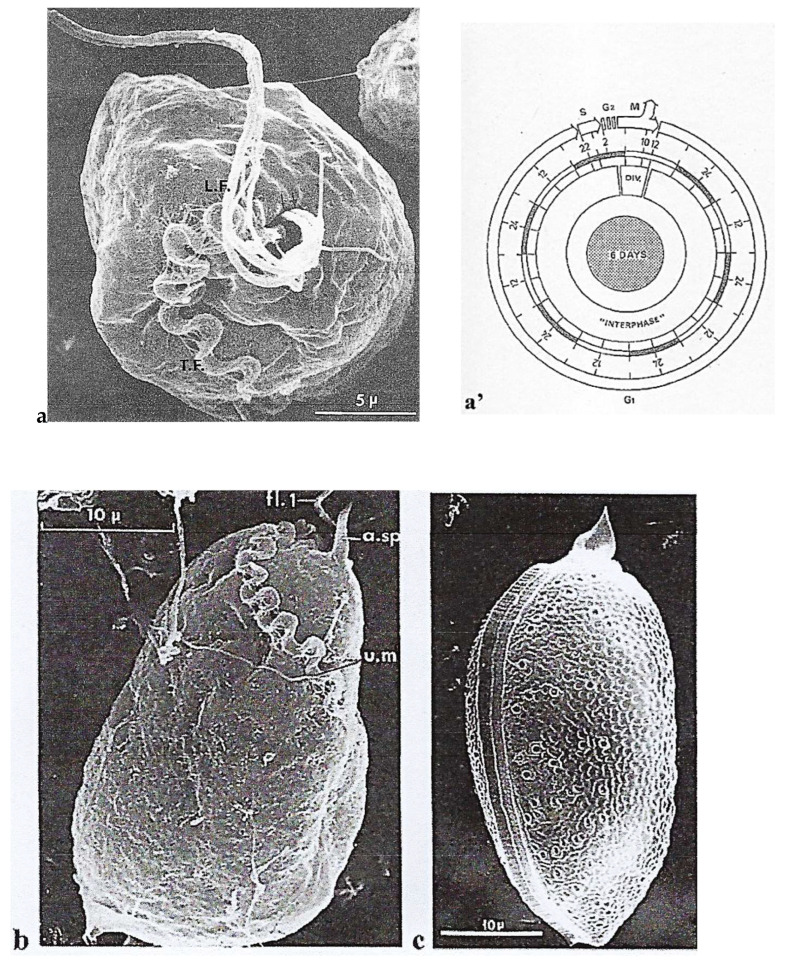
Scanning electron microscope images of the autotrophic dinoflagellate *P. micans* Ehr. (**a**) Apical view showing the peripheral polysaccharidic envelope (epitheca) conserved after soft centrifugation. Two longitudinal flagella (L.F.) of this pre-dividing cell run more or less parallel to each other; the transverse(oblique) flagellum (T.F.) is also visible. Scale bar = 5 µm. (**a’**) *P. micans* cell cycle and duration of the different phases: S < 4 h; G2 short (<4 h), G2 + M = 8 h; S + G2 + M ≤ 12 h; G1 = 120 h. From Bhaud and Soyer-Gobillard [19]. Courtesy of Elsevier. (**b**) One undulating, transverse (oblique) flagellum lined with an undulating membrane (u.m) arises from the same opening as the longitudinal flagella and adheres to the outer layer. Scale bar = 10 µm (a.sp., apical spine). (**a**,**b**) Reproduced from Soyer-Gobillard et al. [20]. Courtesy of Elsevier. (**c**) After stronger centrifugation, two halves with numerous pores, separated by a central suture, are visible, with the apical spine visible at the top. Scale bar = 10 µm. From M.-O. Soyer-Gobillard et al. [21]. Courtesy of Vie Milieu Life & Environment.

**Figure 2 microorganisms-13-00969-f002:**
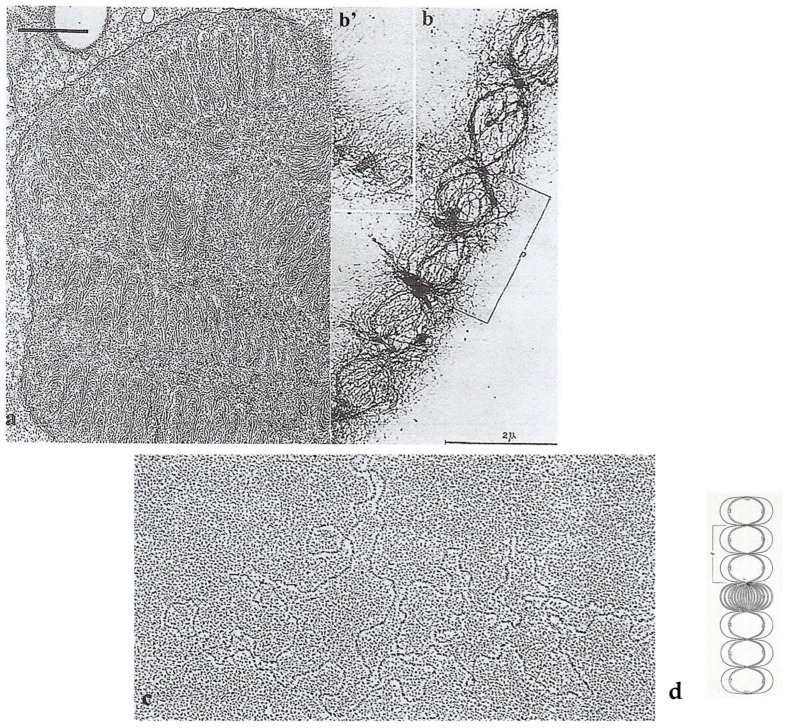
(**a**) TEM observation of an ultrathin section of the nucleus of a non-dividing *P. micans* Ehrenberg cell after specific fixation (Karnovsky-Soyer technique) showing regular arch-shaped chromosomes. Scale bar = 1 µm. From Soyer-(Gobillard) M.-O. [17]. Courtesy of Wiley. (**b**) *P. micans* chromosome spread in water showing the regular unwound periodic organization of the nucleofilaments supporting the hypothesis that *P. micans* chromosome is composed of numerous circular chromatids as shown on the schematic drawing of (**d**) **p**: pitch (around 2 µm). (**b’**) Extrachromosomal filaments. (**c**) Circular chromatid after spreading and carbon-platin shadowing of the molecules from spread extracts of another free-living dinoflagellate, *Crypthecodinium cohnii* Biecheler. In this species, chromosomes are ten times shorter than in *P. micans* Ehr. The length of de DNA molecule is about 120 µm [24]. Unpublished image (M.-O. Soyer-Gobillard). (**d**) Model of circular chromatids illustrating the dinoflagellate *P. micans* chromosomes structure as shown after stretching on water. **p**: pitch (**b**,**b’**,**d**): From Haapala, O.-K. and Soyer-Gobillard M.-O. [22]. Courtesy of Springer Nature.

**Figure 3 microorganisms-13-00969-f003:**
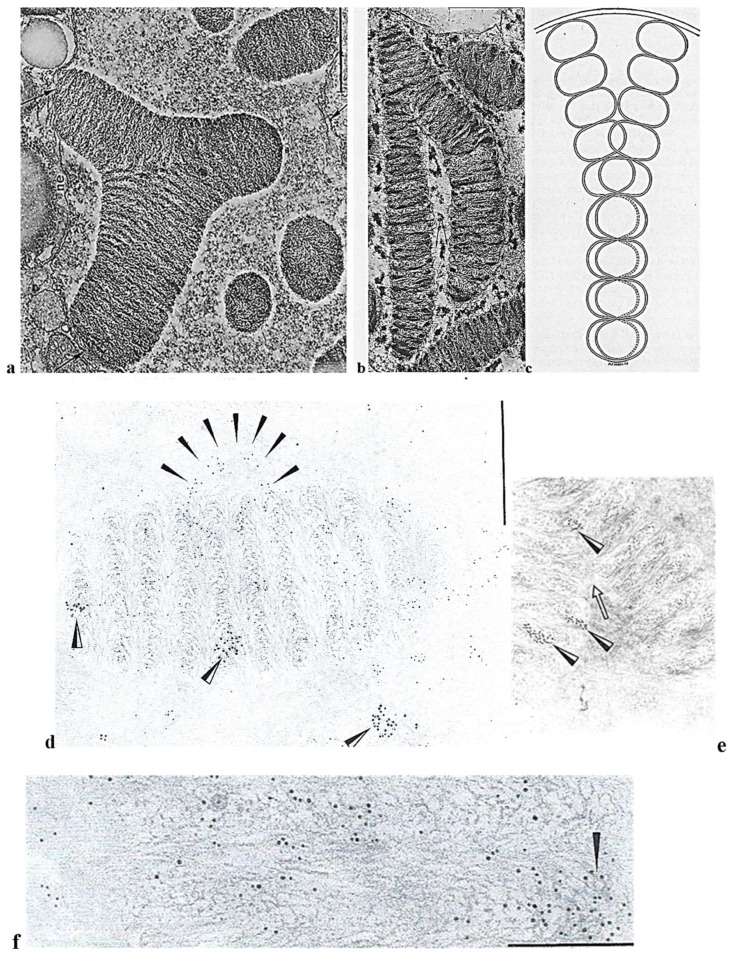
(**a,b**) TEM images of *P. micans* Ehr. dividing chromosomes: (**a**) The ultrathin section of dividing chromosomes fixed to the nuclear envelope (arrows) in which the nucleofilament regular organization can be observed. (**b**) Segregating chromosomes, close to the nuclear envelope (not visible on this picture), supporting the hypothesis that chromosomes are made of many circular chromatids, as shown in the schematic representation of our model (**c**). From Soyer-(Gobillard) M.-O. [21]. Courtesy of Wiley. (**d**) Double labeling of chromosome nucleofilaments with anti-B-DNA antibody (black arrowheads, 5 nm gold particles) forming a loop in the chromosome periphery and Z-DNA antibody (white and black arrowheads); observe the clusters of 7 nm gold particles. Bar = 0.5 µm. (**e**) Immunolocalization of Z-DNA on dividing chromosomes. The white arrow is located in the fission zone. Observe the clusters of 7 nm gold particles (black and white arrow heads). Bar = 0.5 µm. (**f**) High magnification showing immunolocalization of B-DNA with 5 nm gold particles on the chromosome nucleofilaments visible in the background. Bar = 0.2 µm. From Soyer-Gobillard et al. [18]. Courtesy of Rockefeller University Press.

**Figure 4 microorganisms-13-00969-f004:**
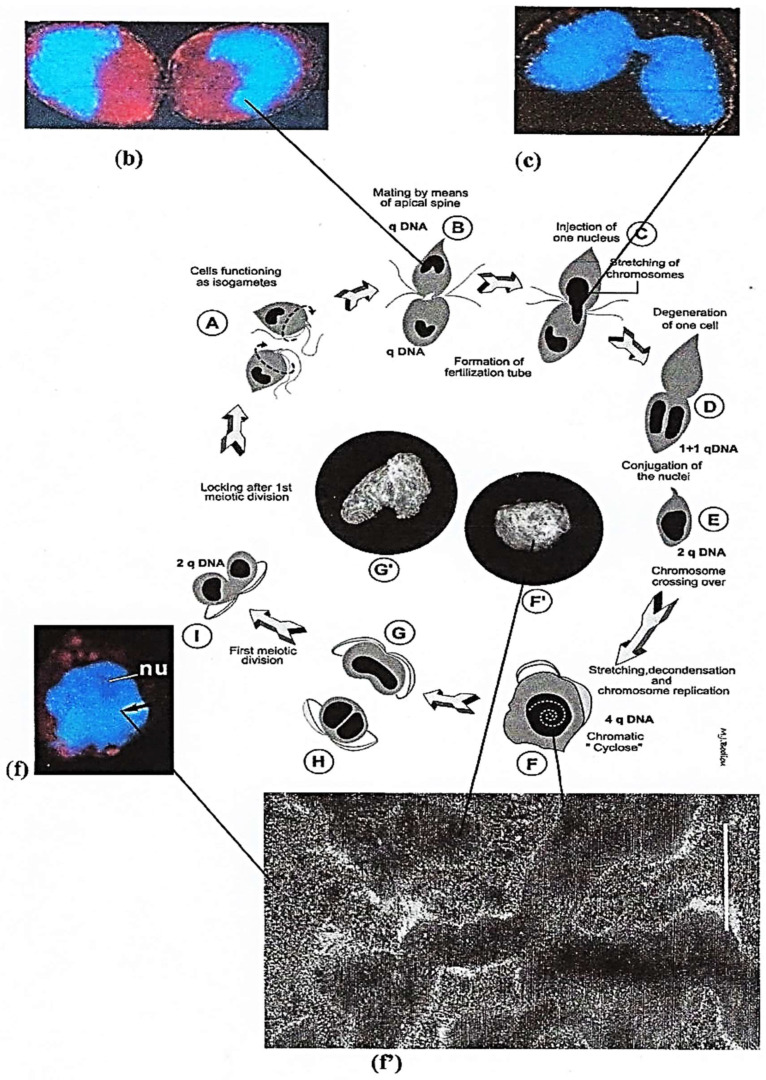
Sexual reproduction phases of the dinoflagellate *P. micans* Ehr. (**A**–**I**) Diagram based on in vivo observations of the nuclei after exposure of cultured *P. micans* cells to low temperature (4 °C) in the dark for 12 h. Vegetative cells, which function as isogametes (**A**) and contain n = qDNA, become paired through their apical spines (**B**). Then, the donor cell injects its nucleus with stretched chromosomes into the receiver cell (**C**). (**b**,**c**,**f**,**f’**) In vivo DAPI-stained nuclei of *P. micans* Ehr cells (blue color): (**b**) corresponds to B, (**c**) to (**C**), (**f**) to (**F**). After the conjugation of the two nuclei (**D**) in the zygote containing n = 2qDNA (**E**), the chromosomes cross over. Chromosomes become completely unwound and stretched (**f’**). In the nucleus containing n = 4qDNA (**F,F’**), chromatin begins to spin around on the right (chromatic cyclosis), giving a round shape to the nucleus (nu) as shown in (**f**). (**f’**) TEM image of the disorganized chromosome structure during chromatic cyclosis corresponding to microscope optical pic-tures (**f**,**F**,**F’**). Scale bar = 1 µm. Only one meiotic division occurs (**G**,**G′**,**H**), leading to n = 2qDNA-containing cells (**I**). From Soyer-Gobillard, M.-O. et al. [29]. By copyright permission from Vie Milieu Life and Environment.

**Figure 5 microorganisms-13-00969-f005:**
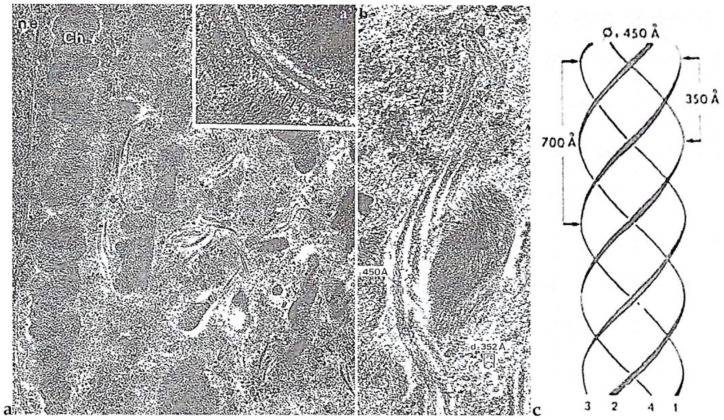
Intranuclear microcables in the nucleoplasm observed after *P. micans* cell vitrification (−269 °C) followed by cryosubstitution according to the methods described in [17]. (**a**) At this meiosis stage, during the chromatic cyclosis, chromosomes (Ch.) are unwound (× 37,000), and (**a’**,**b**), nucleoplasmic microfilaments are supercoiled (black arrowhead) (× 110,000). (**c**) The 450A diameter structure represents the longitudinal section of the greatest measured width of four supercoiled microfilaments organized in microcables. From Soyer-(Gobillard), M.-O. [32]. Courtesy of Elsevier. (**b**,**c**) Unpublished images of M-O. S-G.

**Figure 6 microorganisms-13-00969-f006:**
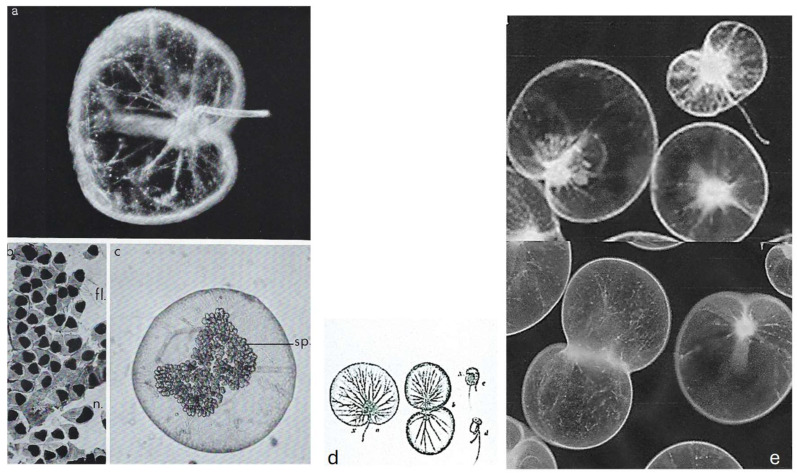
The heterotrophic dinoflagellate *N. scintillans* McCartney. (**a**) Adult trophozoite (diploid) observed in vivo with its tentacle X 65. Photo J. Lecomte, Arago Laboratory. (**b**) Uniflagellated spores (haploid) about to be released. n. nucleus, fl. flagellum. X 1075. (**c**) Overview of *N. scintillans* McCartney about to release its spores. X 65. (**a**–**c**) From M.-O. Soyer-(Gobillard [39]. Courtesy of Springer Nature. (**d**) Schematic representation of a vegetative cell (**a**, left), its binary fission (**b**, middle), and sporocytes (**c**, **d**, right). (**e**) Dividing cells in vivo. (**d**,**e**) https://www.istockphoto.com/fr/vectoriel/image-de-zoologie-de-biologie-antique-noctiluca-miliaris-gm1454192599-489914637 (accessed on 17 April 2025).

**Figure 7 microorganisms-13-00969-f007:**
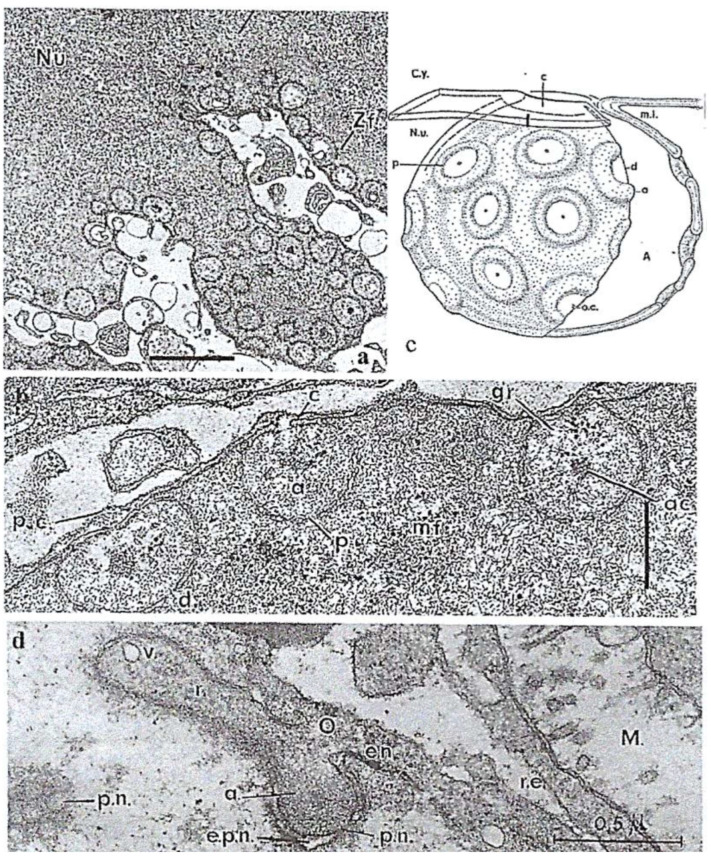
(**a**–**d**) The ultrathin section of the nuclear envelope of the trophozoite of *Noctiluca scintillans* McCartney and the differentiation of its whole internal surface into hundreds of ampullae. (**a**) A partial view of a *N. scintillans* trophozoite nucleus (Nu.) showing the nuclear ampullae filled with nuclear pores, the organization of which constitutes the whole nuclear membrane (Zf. Fibrous zone). Scale bar = 2 µm. (**b**) Higher magnification image showing nuclear ampullae (a) filled with pores (p) that communicate with the cytoplasm through an opening set with a collar (c) and revealing a granular substance (gr) of nucleolar origin (i.e., ribosomes). In the trophozoite, the chromatin is fully decondensed (mf: microfibrils). Scale bar = 1 µm. (**c**) A schematic representation of a nuclear ampulla with its collar and the organization of the nuclear pores. A, bulb; a, ring; c, collar, Cy, cytoplasm; d, diaphragm; m.i., inner membrane; p, pore. (**d**) Fusion of the nuclear membranes at the level of the ampullae (**a**) (stage: 2–4 nuclei). Reproduced from M.-O. Soyer-(Gobillard). [40]. II. Rôle des ampoules nucléaires et de certains constituants cytoplasmiques dans la mécanique mitotique [41]. Courtesy of Wiley, Society of Protistologists.

**Figure 8 microorganisms-13-00969-f008:**
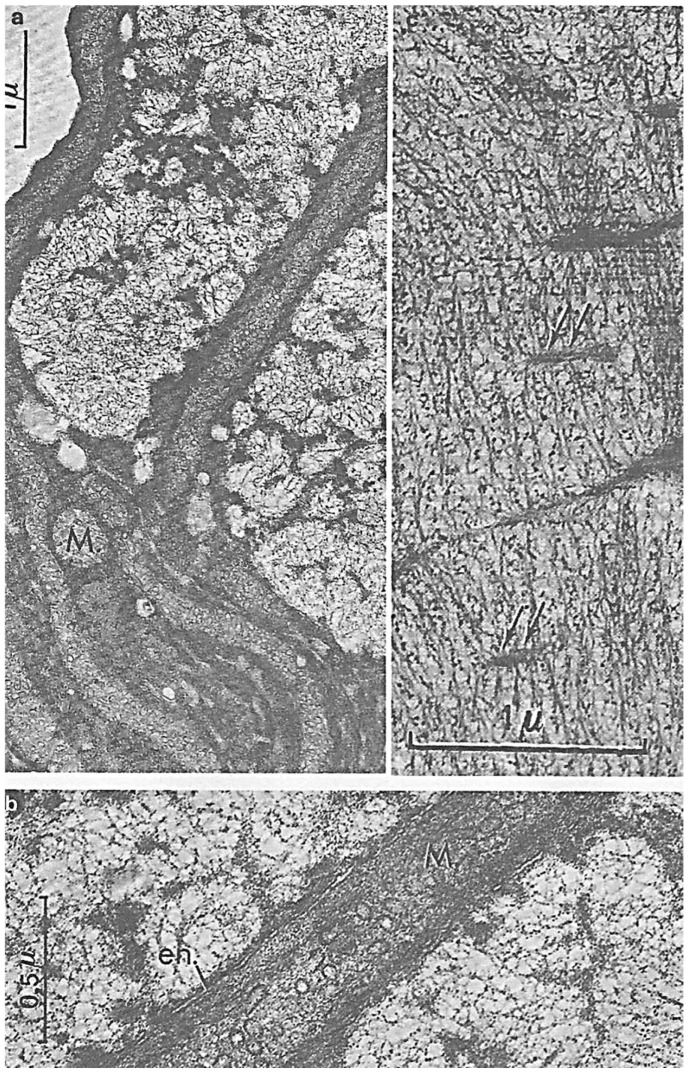
*N. scintillan* McCartney during sporulation. At the first stage of division, the trophozoite chromatin is never organized into chromosomes and the nucleofilament mass will separate into two parts, while giant mitochondria (M) (**a**,**b**) enter the channels bordered by the nuclear envelope (en.). As divisions progress, the chromatin compacts, revealing an organization in a series of arches (**c**) with axial formations (arrows) that initiate the future separation between the chromosome masses. From Soyer-(Gobillard) M.-O. [42]. Courtesy of Springer Nature.

**Figure 9 microorganisms-13-00969-f009:**
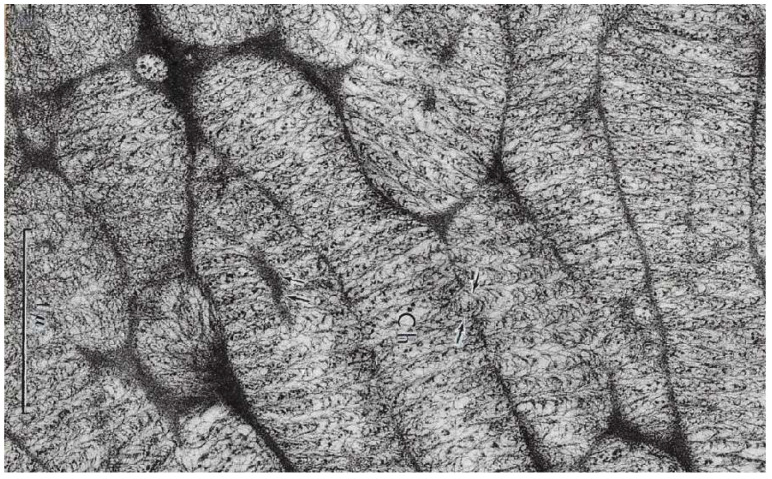
*N. scintillans* McCartney sporulating nucleus (stage: 8 × 2 nuclei) in which the chromatin is being organized into chromosomes with arch-shaped nucleofilaments. Reproduced from M.-O. [42]. Courtesy of Springer Nature.

**Figure 10 microorganisms-13-00969-f010:**
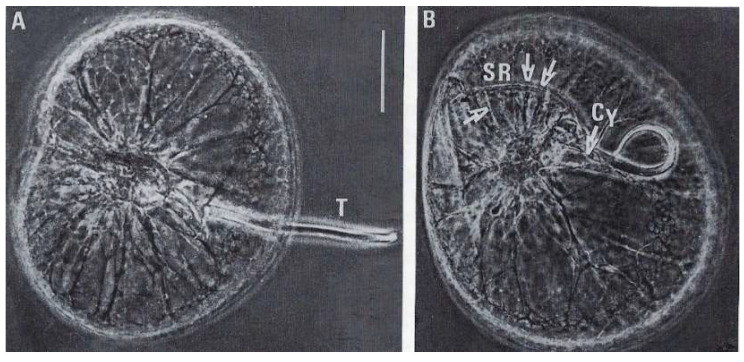
Phase contrast microscopy images of a *N. scintillans* McCartney trophozoite. (**A**) Relaxed tentacle (T). (**B**) Contracted tentacle with its tip close to the cytostome (Cy). Tracts of myonemes (arrows) are anchored between the cytostome and the supporting rod (SR). Scale bar = 200 µm. Photographs by J. Lecomte [43]. Courtesy of Springer Nature.

**Figure 11 microorganisms-13-00969-f011:**
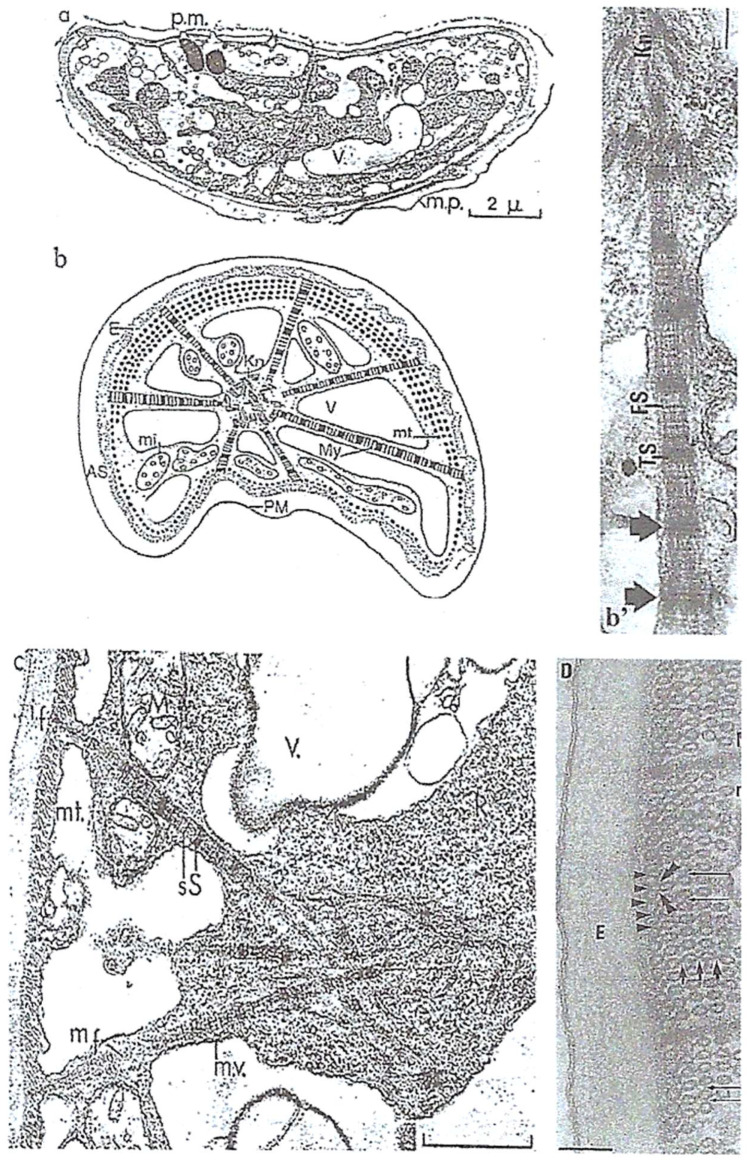
Fine structure of the *N. scintillans* McCartney tentacle. (**a**) TEM image of an ultrathin transverse section of the tentacle showing the striated myonemes inserted in the epiplasm (E) lined with microtubules. (**b**) Schematic drawing of a transverse section of the *N. scintillans* tentacle showing a knot (Kn) of myonemes on the axis, the plasma membrane (PM), vacuoles (V), mitochondria (mi), microtubular rows, and the peripheral alveolar space (AS). (**b’**,**c**) Double striation (S, s, arrows) of striated myofibers organized in myonemes and forming a node in the tentacle axis. **c**: bar = 0.5 µm. (**D**) Higher magnification of a transverse section of the tentacle showing the insertion of myonemes fixed on the epiplasm (E) between several (5) rows of microtubules linked together (arrows). (**a**–**c**) Reproduced from Soyer-(Gobillard) M.-O. [39]. Courtesy of Springer Nature. (**b’,D**) Reproduced from Métivier, Ch. and Soyer-Gobillard, M.-O. [43]. Courtesy of Springer Nature.

**Figure 12 microorganisms-13-00969-f012:**
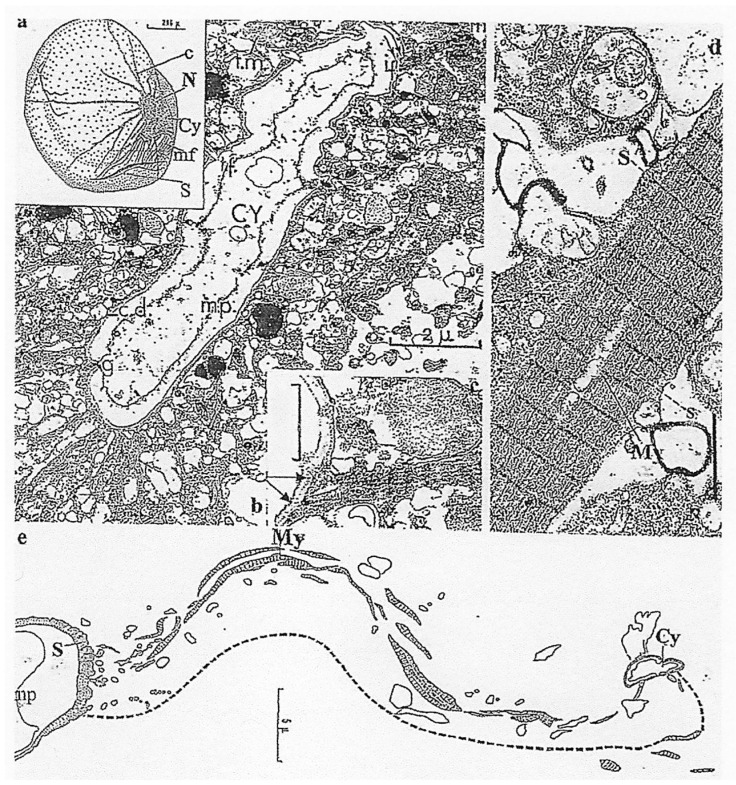
The fine structure of the cytostome (oral apparatus) in *N. scintillans* McCartney. (**a**) A schematic representation of a *N. scintillans*: the cytostome (Cy) is connected to a fixed rod structure or sulcus (S) by a curtain of many myofilaments (mf). N, nucleus, c, cytoplasmic span. (**b**) A transverse section showing the fine structure of the cytostome (Cy) bordered by a thickened lip (arrow). (**c**) Anchoring of striated myonemes (arrows) to the cytostome border. Scale bar = 0.5 µm. (**d**) A striated myoneme (My) composed of many myofibrils. Scale bar = 1 µm. (**e**) A schematic representation of the connection between the anchoring sulcus (S) and the cytostome (Cy) by long ribbons of myonemes (My) mp: plasmic membrane. (**a**–**e**): Reproduced from Soyer-(Gobillard), M.-O. [39]. Courtesy of Springer Nature.

**Figure 13 microorganisms-13-00969-f013:**
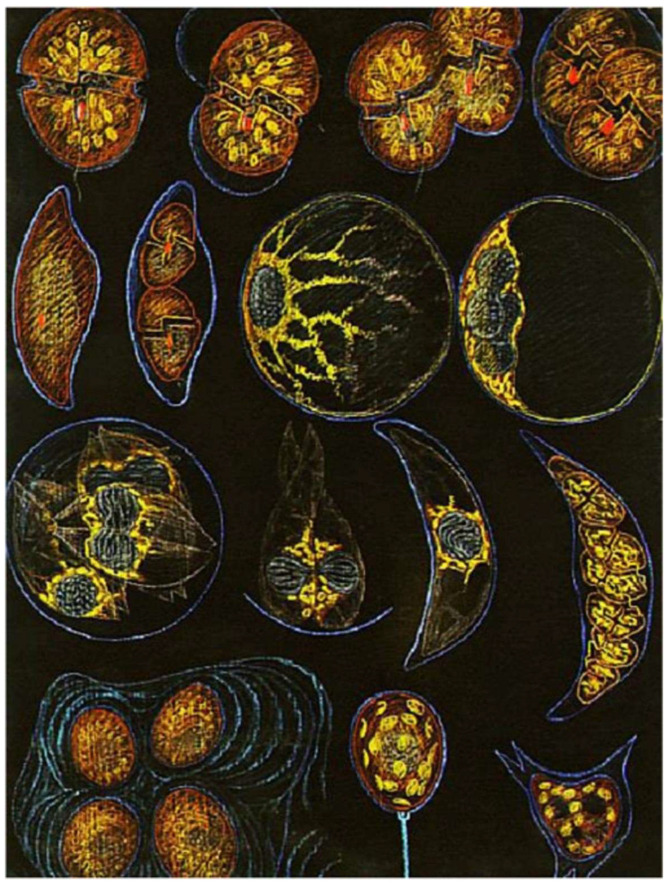
Several free-living bioluminescent (athecate and thecate) dinoflagellates and the life cycle of the genus *Pyrocystis* Murray ex Haeckel, 1890. The species *Pyrocystis pseudonoctiluca* Wyville-Thomson, 1876, is in the second row on the right; *Pyrocystis lunula* Schütt 1896 is in the third row, on the right. On the first row, on the right, is represented *Lingulodinium* spp. J.D. Dodge, 1989, *(Gonyaulax polyedra* F. Stein, 1883), a thecate bioluminescent dinoflagellate. The blue color of the cell cover simulates their bioluminescence at night. In these species, this phenomenon, as well as its chemistry and molecular control, has been described [35,36]. Reproduced from Soyer-(Gobillard), M.-O. and Schrével, J. [3]. Copyright of page 115 is courtesy of “Bibliothèque du Laboratoire Arago-Sorbonne Université”, bequest Lwoff.

**Figure 14 microorganisms-13-00969-f014:**
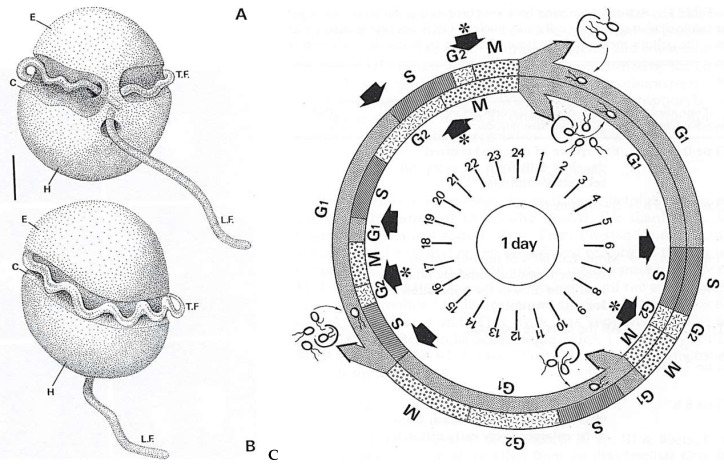
(**A**,**B**) A schematic representation of a *Crypthecodinium cohnii* Biecheler cell. Upper part: Ventral view; lower part: dorsal view. Also indicated are the episome (E), hyposome (H), longitudinal flagellum (LF), transverse flagellum (TF) and cingulum (C). Bar = 5 µm. Reproduced from Perret, E. et al. [48]. Courtesy of Wiley (The Company of Biologists Limited). (**C**) A diagram of two complete successive *C. cohnii* cell cycles (16 h), represented by two circles inside each other. At the end of each, four daughter cells are released. The transition points G1-S (‘start’ point) and G2-M are represented by arrows and by arrows plus an asterisk, respectively. Reproduced from Bhaud, Y. et al. [47]. Courtesy of Wiley (The Company of Biologists Limited).

**Figure 15 microorganisms-13-00969-f015:**
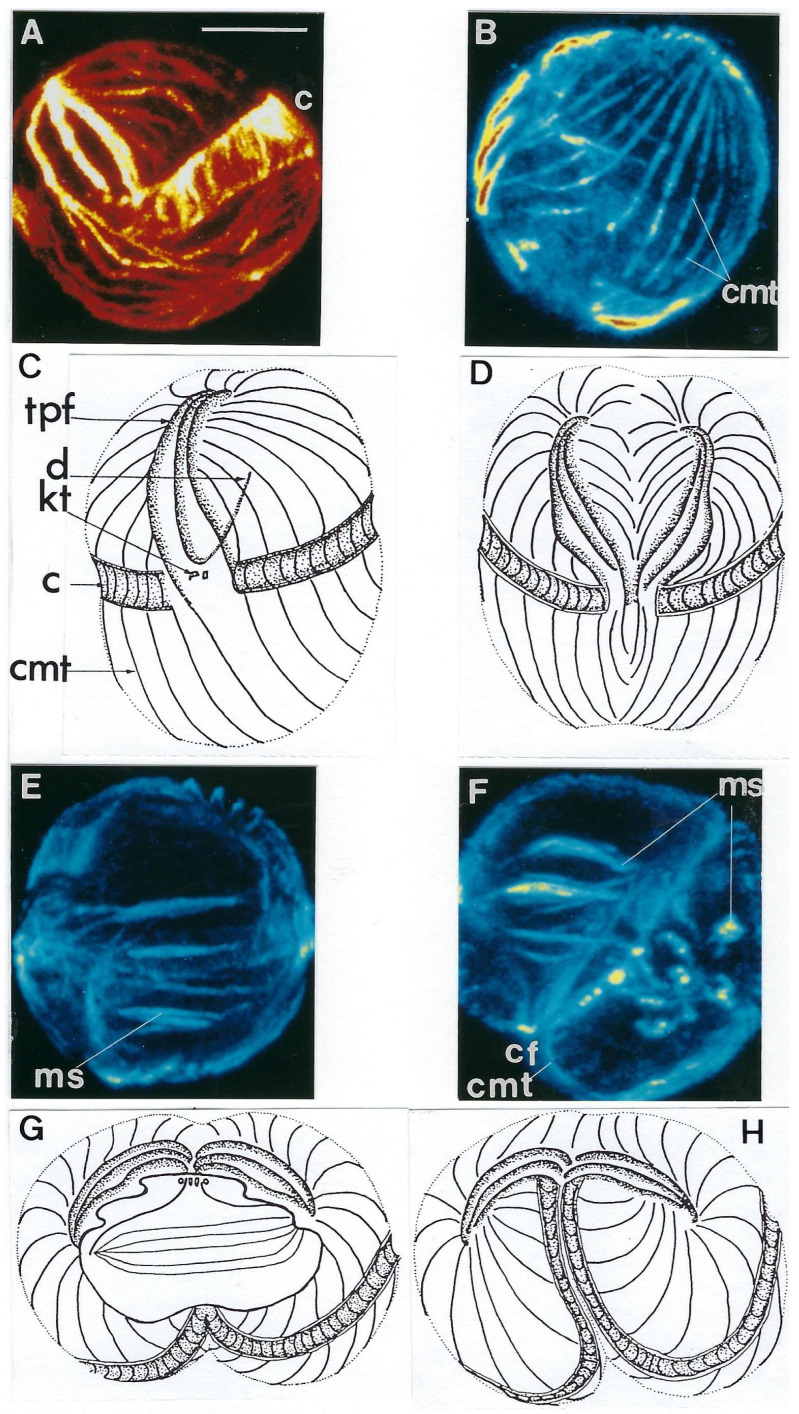
(**A**–**H**) The organization of the cortex microtubules and the microtubular spindle during mitosis as observed by confocal microscopy. Images (**A**,**B**,**E**,**F**) are reconstructions of each 16 confocal laser scanning sections of *C. cohnii* cells after labeling with anti-β-tubulin antibody. Images (**C**,**D**,**G**,**H**) are schematic drawings interpreting the confocal images. Bar = 10 µm. (tpf: three-pronged fork, d: desmose, ms: mitotic spindle, cf: cleavage furrow, c: cingulum, cmt: cortical microtubular rows, kt: kinetosomes. Reproduced from Soyer-Gobillard M.-O. et al. [49]. Courtesy of Elsevier.

**Figure 16 microorganisms-13-00969-f016:**
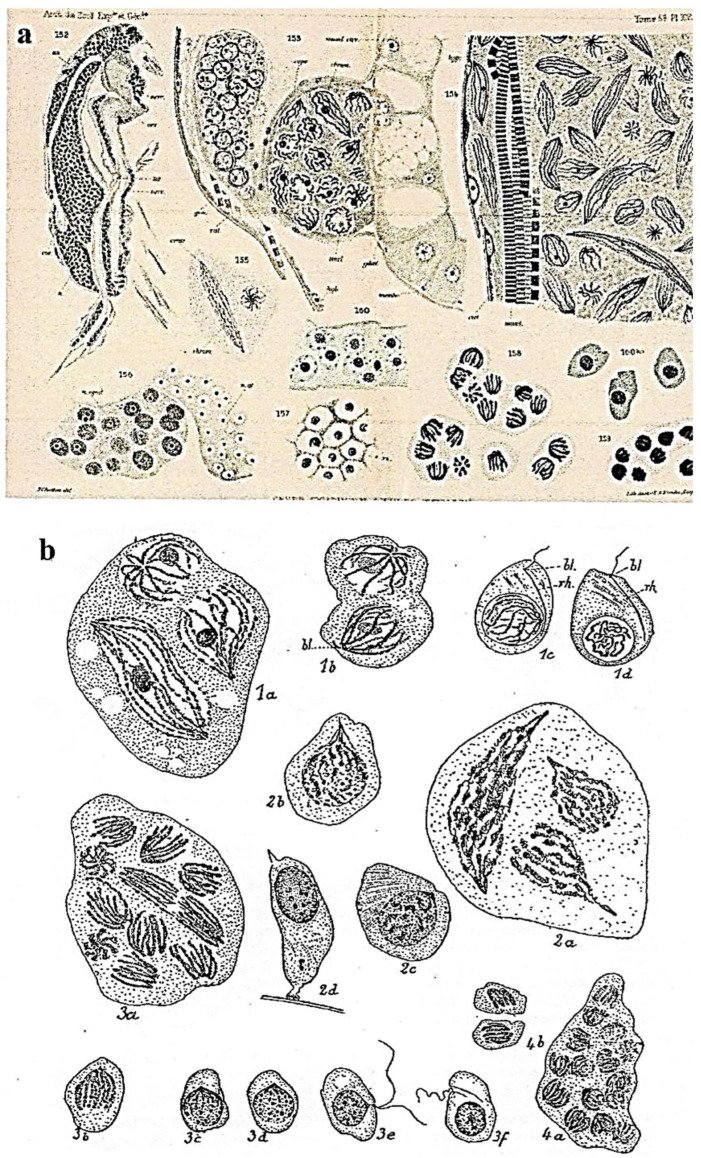
(**a**) *Syndinium turbo* Chatton, coelomic parasite of pelagic copepods. At the top, a plasmodium with nuclei containing five chromosomes. At the bottom, on the right, sporocytes with their five chromosomes. *In* Thèse de CHATTON, E. [1]. (**b**) Plasmodia with dividing nuclei and dinospores (bottom) rh: rhizoplast, bl: blepharoplast (organelles of the flagellar structure). 1. *Syndinium rostratum*; 2. *S. corycoei*; 3. *S. turbo*; 4. *S. microsporum*. *E. Chatton del.* From Titres et Travaux Scientifiques by Chatton [2]. *E. Chatton del*.

**Figure 17 microorganisms-13-00969-f017:**
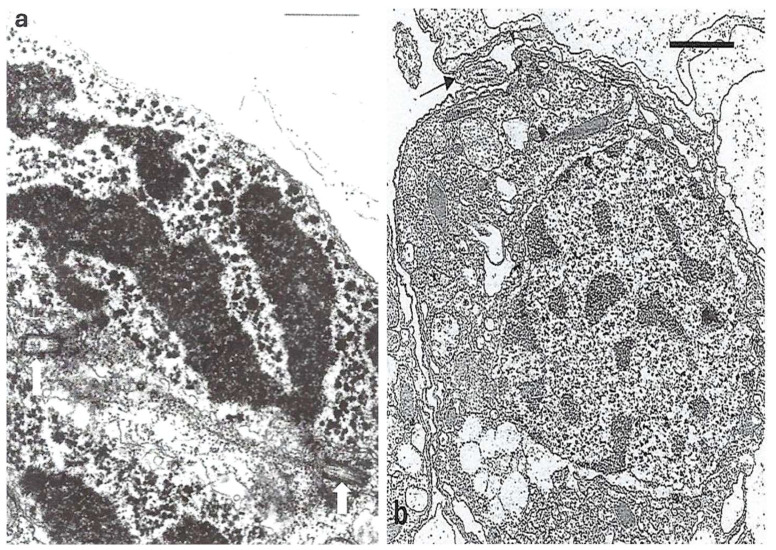
TEM images of *Syndinium* sp. (**a**) One dividing cell showing one of the five V-shaped chromosomes linked by microtubules of the mitotic spindle to one of the two centrioles that compose the centrosome (white arrows). Scale bar = 0.5 µm From Ris, H. and Kubai, D.F. [58]. Courtesy of Rockefeller University Press. (**b**) Sporocyte of *Syndinium sp.* just before its emission from the coelomic cavity of the Copepod. No plastid is visible. The chromosomes are fragmented in the nucleus and an external flagellum is visible in cross section (arrow). Scale bar = 0.5 µm. From Soyer-(Gobillard) M.-O. [59]. Courtesy of *Vie Millieu Life & Environment*.

**Figure 18 microorganisms-13-00969-f018:**
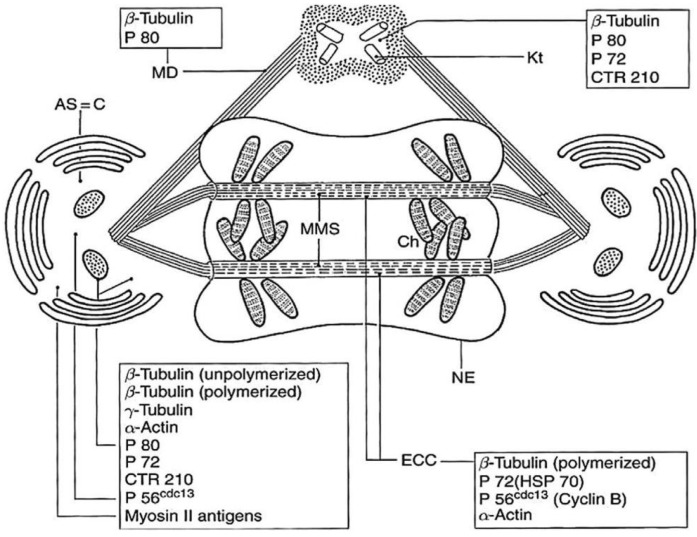
A schematic representation of a dinoflagellate (except *Syndinium*, *Oxyrrhis* and, probably, *Noctiluca*) mitotic apparatus in anaphase with the list of remarkable centrosome-associated proteins. Microtubular mitotic spindles lie throughout the nucleus, pass into the archoplasmic spheres (Golgi apparatus), and are linked to the two pairs of kinetosomes or flagellar bases. AS, archoplasmic sphere (containing Golgi bodies); C, centrosome (without centrioles); ECC, extranuclear cytoplasmic channel; MMS, microtubular mitotic spindle; MD, microtubular desmose; Kt, kinetosomes; N, nucleus; NE, nuclear envelope (permanent); Ch, chromosomes. Reproduced from Soyer-Gobillard, M.-O. et al. [49]. Courtesy of Elsevier.

**Figure 19 microorganisms-13-00969-f019:**
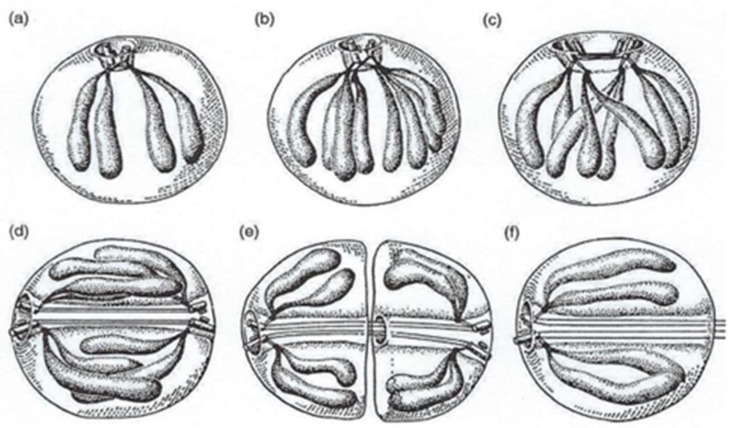
A schematic representation of nuclear division of *Syndinium* sp. that contains five V-shape chromosomes. (**a**) Interphase. (**b**) Early division: centrosome (two centrioles), kinetochores and chromosomes have duplicated. (**c**) An early stage of chromosome segregation. The central spindle between separating centrioles. (**d**) A late stage of chromosome segregation. The central spindle in cytoplasmic channel throughout the nucleus. (**e**) Division of nucleus. (**f**) An early daughter nucleus with persisting channel and microtubules. Reproduced from Ris, H. and Kubai, D.F. [58]. Courtesy of Rockefeller University Press.

**Figure 20 microorganisms-13-00969-f020:**
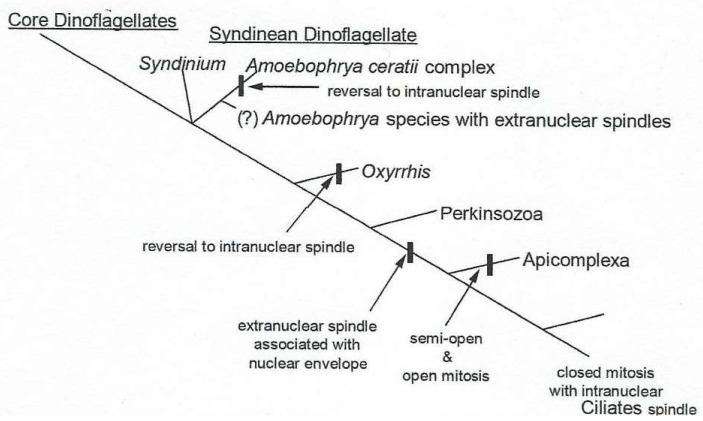
The evolution of the mitotic apparatus in alveolates according to the phylogeny by Bachvaroff et al. [7] and reproduced from Moon, E., et al. [60]. Courtesy of Elsevier.

**Figure 21 microorganisms-13-00969-f021:**
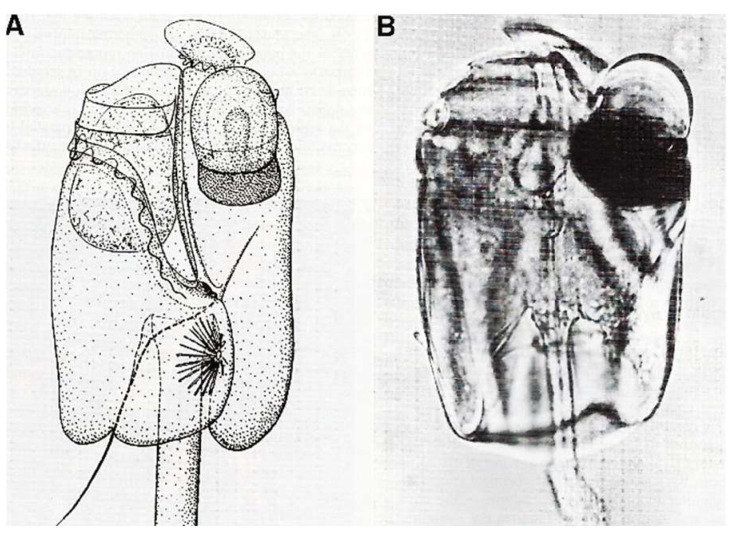
Ocelloid of the dinoflagellate *Erythropsis pavillardi* Hertwig. (**A**) A schematic drawing of *E. pavillardi* in which the eyespot is represented on the right of the cell anterior part. (**B**) Light microscopy image showing the hyalosome, which plays the role of the lens (upper part), and the pigment, which plays the role of the retina. From Greuet C. [61].

**Figure 22 microorganisms-13-00969-f022:**
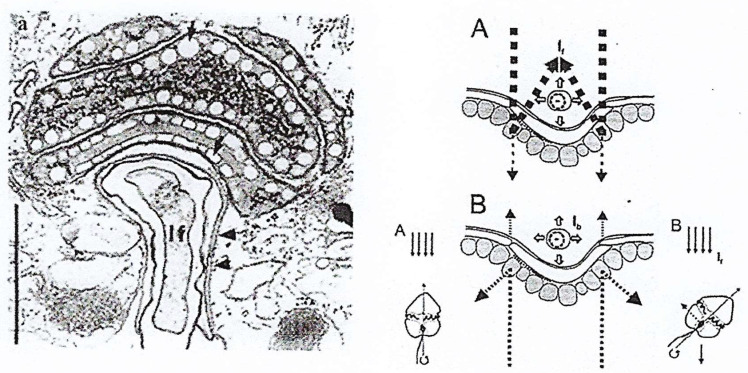
TEM images of the eyespot of the binucleated dinoflagellate *G. foliaceum* in the transverse section (**a**) localized in the posterior part of the cell. The images show the layers of refracting globules that form the pigment cup, composed of carotenoid-rich lipids through which light passes (**A**) and/or is reflected (**B**), which determines the cell orientation, according to Kreimer’s hypothesis. Dotted arrows indicate the direction of the light reflected or passing through the cell. The longitudinal flagellum (lf) is visible in the hollow of the posterior sulcus. Bar = 0.5 µm. Reproduced from Kreimer G. (with permission). [69]. Courtesy of Elsevier.

**Figure 23 microorganisms-13-00969-f023:**
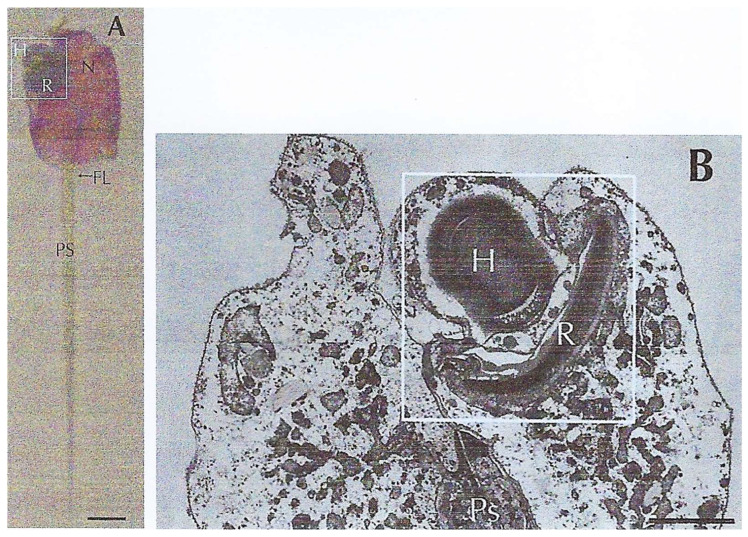
*Erythropsidinium* spp. dinoflagellate equipped with a piston (PS) involved in its locomotion in water. (**A**) Light microscopy view of *Erythropsidinium* spp. with the ocelloid in the left anterior part of the cell (square). PS: piston; R: retina-like body; H: hyalosome; Fl: flagellum. Scale bar = 20 µm. (**B**) TEM view of the ocelloid. The hyalosome (H), (crystalline body) plays the role of a lens; observe at its base the lamellated retina-like body (R). Scale bar = 10 µm. From Hayakawa, S. et al. [72]. Courtesy of *PLoS ONE*.

**Figure 24 microorganisms-13-00969-f024:**
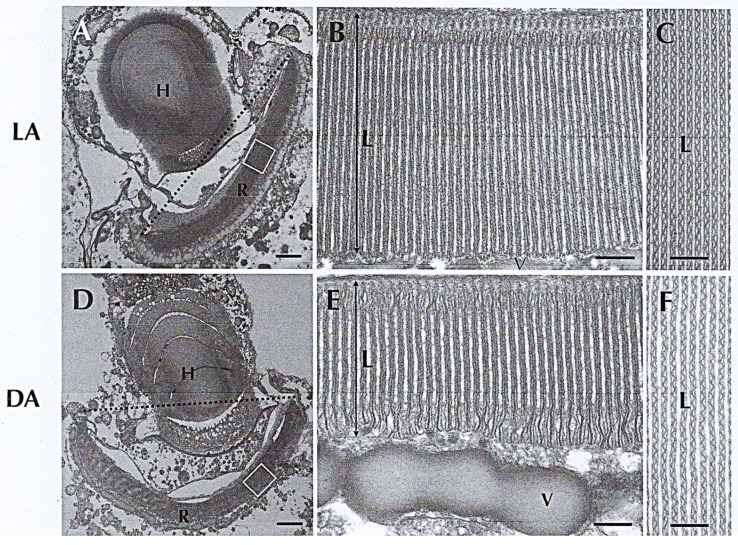
(**A**–**F**). Detail of ocelloid of *Erythropsidinium* spp. and the variations of its morphology in different conditions of light. Light state (**LA**: **A**–**C**) and dark state (**DA**: **D**–**F**). (**B**,**E**): longitudinal sections of retinal body; (**C**,**F**). cross sections of retinal body are shown. **L**. lamellae; **V**. vesicular layer. **A**,**D**. Bar = 2 µm; **B**,**C**,**E**,**F**, Bar= 0.2 µm. From Hayakawa, S. et al. [72]. Courtesy of *PLoS ONE*.

## Data Availability

No new data were created or analyzed in this study. Data sharing is not applicable to this article.

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
