# Peer review of "Some Insights into the Inventiveness of Dinoflagellates: Coming Back to the Cell Biology of These Protists"

_microorganisms, 2025, doi:10.3390/microorganisms13050969_

Round 1
Reviewer 1 Report
Comments and Suggestions for Authors
This is an interesting retrospective by Prof. Soyer-Gobillard, and it reflects her personal contributions and interests. The MS is interesting and worth publishing. My major critique is that some of the figures are difficult to read, and some of the labels on the panels are not explained in the legend.
I have a number of smaller points which would help the MS if addressed:
Line 90 Some definitions for terminology might be helpful. Maybe a schematic picture of a cell with different terminology used in the text indicated?
Line 95 LF hard to see in the image. There also seems to be an IF near the top?
Line 101 Some things labeled in image that are not in the legend
Line 105 is the vertical line like structure a suture or a spine?
Line 110 “The nucleus is filled with UP TO A hundred large chromosomes (10 μm long and 1 μm wide) 110 (Figure 2a) and IS surrounded by large plastids”
Line 120 “P. micans has two types of reproduction: a vegetative reproduction, carried out by binary fission, and a sexual reproduction.”
Line 126 I’m not sure what this means, centromeres and telomeres are very different in structure and function. Also, if the chromosomes are circular, why would there be telomeres at all?
Line 144 A nice picture of a circular DNA molecule. However, I calculate less than 500,000 base pairs in a 120 um circle. Would this not be a bit small for a nuclear chromosome? And too big for a plastid minicircle?
Line 151 Is the nuclear membrane visible in panel b?
Line 196 Is there a panel (a)?
Line 202 Does one nuclear membrane fuse with the other to put homologous chromosomes in contact?
Line 205 what is chromatic cyclosis?
Line 206 what is 2q? Should it be 2n? 2c?
Line 224 the 450 A in picture, does it refer to the diameter of the filaments? If so they would be about twice that of microtubules.
Line 242 define amphiesma? Pellicule? (see comment for line 90)
Line 248 is the ploidy of spores the same as the trophozoite?
Line 250 scale bars?
Line 261 are ampullae unique to N.scintillans?
Line 267 is 7d the right picture to show the smooth nuclear envelope?
Line 268 Trophocyte?
Line 273 Nu and Zf in panel a?
Line 280 I was not aware that the “permanently condensed” chromosomes were ever decondensed in normal vegetative cells (trophozoites). Is it unique to N. scintillans? Or is this something do with spore formation?
Line 290 is this still called a trophozoite?
Line 304 cytostome? Myonemes?
Line 327 I don’t see the arrows in panel c. Is this meant to be panel d?
Line 357 First row looks like a Lingulodinium. If they are, they do have a theca.
Line 391 I do not see E or H in the image.
Line 395 Why are there two different circles (inner and outer) in panel b?
Line 408 what is a plasmodium? Should maybe explain as most people will probably only be familiar with Plasmodium
Line 478 I am not sure that centrioles “govern mitosis”. They are absent in plants.
Line 535 Both A and B appear to show light being reflected.
Author Response
- Answers to Reviewer 1 : Most of the improvements were surlined in yellow in the MS.
This is an interesting retrospective by Prof. Soyer-Gobillard, and it reflects her personal contributions and interests. The MS is interesting and worth publishing. My major critique is that some of the figures are difficult to read, and some of the labels on the panels are not explained in the legend.
I have a number of smaller points which would help the MS if addressed:
Line 90 Some definitions for terminology might be helpful. Maybe a schematic picture of a cell with different terminology used in the text indicated?
I agree, so I have completed as well as possible all the explanations in the text.
Line 95 LF hard to see in the image. There also seems to be an IF near the top?
I agree. I have changed the Figure a and added letters more visible L.F. and T.F. on the figure and on the legend of fig a.
Line 101 Some things labeled in image that are not in the legend
I agree. I have completed the legend.
Line 105 is the vertical line like structure a suture or a spine?
I have completed the legend : spine is visible at the top.
Line 110 “The nucleus is filled with UP TO A hundred large chromosomes (10 μm long and 1 μm wide) 110 (Figure 2a) and IS surrounded by large plastids”
I agree : « is surrounded by large plastids, not visible on this nucleus picture. »
Line 120 “P. micans has two types of reproduction : a vegetative reproduction, carried out by binary fission, and a sexual reproduction.”
I agree : «P.micans has two types of reproductions :a vegetative reproduction, carried out by binary fission, and a sexual one ».
Line 126 I’m not sure what this means, centromeres and telomeres are very different in structure and function. Also, if the chromosomes are circular, why would there be telomeres at all?
I agree: I have explicited the sentence: “In these rod-shaped chromosomes, centromeres could be localized at the chromosome tip and they can be considered to be (virtual) telomeres. However, centromeric heterochromatin being lacking, P. micans could be considered as a kind of “immortal cell”, unable to accomplish apoptosis “.
Line 144 A nice picture of a circular DNA molecule. However, I calculate less than 500,000 base pairs in a 120 um circle. Would this not be a bit small for a nuclear chromosome? And too big for a plastid minicircle?
Yes I agree : Indeed, this circular DNA molecule was spread not from P. micans extracts but from another species, Crypthecodinium (Gyrodinium) cohnii which presents chromosomes about ten times shorter than P. micans.
So, I corrected the legend : « Circular chromatid after spreading and carbon-platin shadowing of a DNA molecule from spread extract of another free-living dinoflagellate, Crypthecodinium cohnii Biecheler. In this species, chromosomes are about ten times shorter than in P. micans Ehr. The length is about 120 µm ».
Line 151 Is the nuclear membrane visible in panel b?
I have modified the sentence in the legend :
« Segregating chromosome, close to the nuclear envelope (not visible on this picture) and supporting the hypothesis that chromosomes are made of many circular chromatids,… »
Line 196 Is there a panel (a)? Line 202 Does one nuclear membrane fuse with the other to put homologous chromosomes in contact? Line 205 what is chromatic cyclosis? Line 206 what is 2q? Should it be 2n? 2c?
To answer to these 4 points I have rewritten the following sentences :
« P. micans sexual reproduction was observed in our laboratory by chance, after a glass bottle with some cultured cells was placed in a refrigerator at 4° in the dark for 12 hours. The next morning, many vegetative cells (n=1qDNA) had paired up, clinging one to the other through their apical spine and emitting a connecting tube (i.e. fertilization tube) (Figure 4, A, B, C) through which one cell (male?) injects its genetic material surrounded with nuclear envelope into the other (the female?) (Figure 4, D, E) [29]. Shortly after the fusion of the two nuclear envelopes and nuclei, their chromatin undergoes a very impressive circular and rapid movement at right as observed with phase contrast optical microscope (i.e. chromatic cyclosis) for several minutes, during which the male and female genetic materials mingle and chromosomes lose completely their regular structure (Figure 4, F, f, f’). By quantifying DNA in single cells, we showed that in P. micans, early planozygotes (Figure 4, f) (n=2qDNA content) double their DNA to n=4qDNA before the first of the two zygotic divisions (meiosis), which leads to n=1qDNA in vegetative cells»
Line 224 the 450 A in picture, does it refer to the diameter of the filaments? If so they would be about twice that of microtubules.
Answer : Modified sentence : "The 450 Å diameter structure represents the longitudinal section of the greatest measured width of supercoiled microfilaments organized in microcables." I have added an explicative schema. (Fig (c))
Line 242 define amphiesma? Pellicule? (see comment for line 90)
Answer : I modified the sentence :
« Indeed, its cell covering or “amphiesma” presents an outer membrane that surrounds the cell, amphiesmal vesicles and a thin pellicular layer,.. »
Line 248 is the ploidy of spores the same as the trophozoite?
Answer : It is explained in the legend : the trophozoit is diploid and the spores are haploid.
Line 250 scale bars?
Answer : the magnifications are indicated in the legend.
Line 261 are ampullae unique to N.scintillans?
Answer : I have modified the legend of Fig 7 :« Ultra-thin section of the nuclear envelope of Noctiluca scintillans McCartney and the differentiation of its whole internal surface into hundreds of ampullae. »
Line 267 is 7d the right picture to show the smooth nuclear envelope?
Answer : I agree ; it is Figure 8b.
Line 268 Trophocyte?
I agree : I changed to « Trophozoite »
Line 273 Nu and Zf in panel a?
Answer : I have added in the legend : trophozoite nucleus (Nu.) ; Zf. Fibrous zone
Line 280 I was not aware that the “permanently condensed” chromosomes were ever decondensed in normal vegetative cells (trophozoites). Is it unique to N. scintillans? Or is this something do with spore formation?
Answer : The trophozoite of Noctiluca has never condensed chromatin. As shown in our previous observations, chromatin began to condense to form structured chromosomes during sporocyte formation at stage minimum 8X2=16 nuclei.
Line 290 is this still called a trophozoite?
Answer : The trophozoite has one nucleus. It is vegetative and not dividing. When sporulation is triggered for physicochemical reasons not yet specified in the surrounding environment, multiplications are very rapid by splitting of the chromatic mass then towards stage 16 nuclei begin to condense in individualised.
Line 304 cytostome? Myonemes?
Answer : oral apparatus (cytostome) : I completed the sentence as : « They are located in the tentacle (Figure 10, a, b) and at the level of the oral apparatus (cytostome) where cytoplasmic myofibrils are organized in striated and contractile muscular fibers named myonemes (Fig. 11a, b, b’). »
Line 327 I don’t see the arrows in panel c. Is this meant to be panel d?
I agree : I modified the sentence as : (D) Higher magnification of a transverse section of the tentacle showing the insertion of myonemes fixed on the epiplasm (E) between several (5) rows of microtubules linked together (arrows).
Line 357 First row looks like a Lingulodinium. If they are, they do have a theca.
Answer : You are right ! I added a sentence to the legend : « Several free-living bioluminescent (athecate and thecate) dinoflagellates ….On the first row, at right, is represented Lingulodinium spp. J.D. Dodge, 1989, (Gonyaulax polyedra F. Stein, 1883), a thecate bioluminescent dinoflagellate. »
Line 391 I do not see E or H in the image.
Answer : Indeed, I have redid completely the figure and separated it into 2. (Figures 14 and 15 ).
Line 395 Why are there two different circles (inner and outer) in panel b?
Answer : These two different circles (inner and outer) represent the diagram of two successive C. cohnii cell cycles over 24 h. (explained in the legend) I redid the legend : « (C) Diagram of two complete successive C. cohnii cell cycles over 24 h (=16h), represented by two circles inside each other. At the end of each, are released four daughter cells. The transition points G1-S (‘start’ point) and G2-M are represented by arrows and by arrows plus asterisk, respectively. »
Line 408 what is a plasmodium? Should maybe explain as most people will probably only be familiar with Plasmodium.
I agree : so I redid the sentence : « The “plasmodium”, which is composed of a cytoplasmic mass with numerous nuclei containing five V-shaped chromosomes and without cellular partitioning, is constantly in mitosis, growing in the general cavity (coelomic cavity) of copepods and other crustaceans, and rapidly destroying all their vital organs. »
Line 478 I am not sure that centrioles “govern mitosis”. They are absent in plants.
I agree and removed these words.
Line 535 Both A and B appear to show light being reflected.
I don’t agree : In B (from Kreimer), the light is shown reflected against the refracting globules while in A the light passes through them. (direction of arrows).
Reviewer 2 Report
Comments and Suggestions for Authors
This review is an excellent summary of the results of long-standing studies of biology, life cycle, and ultrastructural organisation of evolutionarily ancient dinoflagellates, with a focus on their amazing features. A survey of the extant literature reveals a paucity of surveys on the fine structure of dinoflagellates, and this manuscript, to my best knowledge, is the first to address issues on cytology of these unicellular organisms in such a comprehensive manner and new. Undoubtedly, the review will nicely contribute to the special issue “Research on Biology of Dinoflagellates” of the journal Microorganisms and will help the readers to evaluate the complexity of life style of and master functions of dinoflagellates. Due to the high quality of writing, the depth of the analysis of the experimental data, and the excellence of the excellent illustrations and images, this manuscript meets all the standards for publication in its current form. The author may wish to consider the following minor points.
Minor comments
In my opinion, it would be beneficial to incorporate additional information (LL245-6) on the role of mature spores (sporocytes). Do they ensure survival and resistance to growth-unfavorable or stress conditions?
In the Conclusions it would be beneficial to outline the issues that need resolution and the methodology to be employed (After L 578).
A technical comment
Some parts in the references are underlined and should be adjusted to the common style.
Author Response
Reviewer 2 :
Comments and Suggestions for Authors
Il est indéniable que votre revue met en lumière la complexité et la beauté de l’organisation ultrastructurale des dinoflagellés, et s’impose comme une œuvre d’une remarquable alliant science et art. Je suis convaincue que cette étude pourra être publiée en l’état, à l’exception de quelques ajustements mineurs dans la liste des références (sans soulignement).
Merci pour votre excellent travail.
This review is an excellent summary of the results of long-standing studies of biology, life cycle, and ultrastructural organisation of evolutionarily ancient dinoflagellates, with a focus on their amazing features. A survey of the extant literature reveals a paucity of surveys on the fine structure of dinoflagellates, and this manuscript, to my best knowledge, is the first to address issues on cytology of these unicellular organisms in such a comprehensive manner and new. Undoubtedly, the review will nicely contribute to the special issue “Research on Biology of Dinoflagellates” of the journal Microorganisms and will help the readers to evaluate the complexity of life style of and master functions of dinoflagellates. Due to the high quality of writing, the depth of the analysis of the experimental data, and the excellence of the excellent illustrations and images, this manuscript meets all the standards for publication in its current form. The author may wish to consider the following minor points.
Minor comments :
In my opinion, it would be beneficial to incorporate additional information (LL245-6) on the role of mature spores (sporocytes). Do they ensure survival and resistance to growth-unfavorable or stress conditions?:
I agree and completed the sentence : « ..and then uniflagellate spores which will ensure the dissemination and resistance to growth-unfavorable conditions. » (LL247-248)
In the Conclusions it would be beneficial to outline the issues that need resolution and the methodology to be employed (After L 578). :
I agree and completed the sentence : « Much research using the most advanced techniques in cell biology is still be necessary to try to solve these enigmas. » (LL 583-584).
A technical comment :
Some parts in the references are underlined and should be adjusted to the common style :
I agree and modified it in the text.